# The Comparability of Financial Information in Insurance Companies Using NiCE Qualitative Characteristics Measurement

Magdalena Chmielowiec-Lewczuk [1,*], Marzanna Lament [2], Kinga Bauer [3] and Ewa Spigarska [4]

1 Department of Insurance, Wroclaw University of Economics and Business, 53-345 Wroclaw, Poland
2 Department of Finance, Insurance and Accounting, Casimir Pulaski Radom University, 26-600 Radom, Poland; m.lament@uthrad.pl
3 Department of Accounting, Cracow University of Economics, 31-510 Cracow, Poland; kinga.bauer@uek.krakow.pl
4 Department of Accounting, University of Gdansk, 80-309 Gdańsk, Poland; ewa.spigarska@ug.edu.pl
* Correspondence: magdalena.chmielowiec-lewczuk@ue.wroc.pl

**Abstract:** The purpose of this article is to assess the comparability of financial information presented in the annual statements of insurance companies by means of the NiCE (Nijmegen Centre for Economics) index of financial report quality assessment and suggest some directions for changes to the assessment of comparability of insurance companies' financial reporting. The selection of cases for the research sample was intentional. Financial reports were chosen from 8 insurance companies, whose share measured by the value of their assets accounts for more than 30% of the EU market. Financial statements for three years (2019, 2020, 2021) were obtained and assessed in each case. The NiCE index of financial report quality assessment was used, and utilised first to assess the qualitative characteristics of financial reporting from entities other than financial institutions. The study found a high comparability in insurance companies' financial reporting. It was also established that the method of assessment is not free from defects, and some improvements were suggested. The results could serve insurance company stakeholders by indicating the current state and some directions for change regarding the comparability of financial statements. The stakeholders require reliable data, mainly regarding the goals on Agenda 2030. Understanding and analysing sustainability goals for entities such as insurance companies without analysing their financial situation is impossible. This research improves the state of the art in the assessment of financial reporting quality and fills a gap in the verification of the comparability of insurance companies' financial information. The research undertaken should also be considered important from the point of view of sustainability, as the quality of information is an important element in decision-making and forms the basis for the preparation of non-financial information and ESG (environment, social, governance) reports.

**Keywords:** insurance companies; financial reporting; comparability of financial statements; qualitative characteristics of financial statements; NiCE index

## 1. Introduction

Information is one of the important factors in any activity, including economic activity, and represents an important value in the process of social, economic, and environmental development, which should be sustainable. The pursuit of sustainable development goals requires information of an adequate level of quality to ensure optimal decision-making and as a tool for communication between decision-makers and the public. An important quality characteristic of information is its comparability.

The comparability of information is expected to enhance its utility and faithful representation of a business and its financial standing. Information is treated as more useful if it can be compared to other, similar information. The similarity of information refers to information offered in the financial reporting both of various entities and of a single entity drafted over different periods of time. Comparability is a highly desirable characteristic

of reliable reporting information, also under the principle of continuity that prevails in accounting. Comparability can also reduce the cost of obtaining information and make it more accessible [1]. Research into the comparability of financial reporting normally relates to the impact of reporting standards on the new ways of presenting financial information, e.g., [2–6], and confirms improvements to its quality. Existing research has covered business groups in a variety of industries, though chiefly non-financial enterprises. It has not regarded insurance companies, an important class of entities from the viewpoint of financial information users. They are financial institutions subject to financial supervision and should provide reliable and comparable data. Due to the number of legal regulations, they are subject to extensive reporting duties (statutory, statistical, solvency, non-financial information—ESG (environmental, social, government)); insurance companies are a homogeneous group of entities whose quality of financial reporting meets the highest standards. It should be noted, though, that these very often global entities operate in diverse conditions and abide by different regulations depending on both their corporate level and location of business. They also vary in the scopes (life and non-life insurance) and scales of their activities. In spite of the rather consistent regulations, this means insurance companies are a heterogeneous group of entities that function in varied conditions, and the information they present may differ.

The issue of comparability of financial information in insurance companies is important to its utility, as it provides the foundation for a variety of assessments and management decisions, both operational and investment. Moreover, insurance companies are so-called public trust institutions, and attention to their image and financial standing are of the essence. The utility and particularly the comparability of information is undoubtedly one of the key characteristics whose realisation helps care for and maintain the good image of these public trust institutions. The research undertaken should also be considered important from the point of view of sustainability, as the quality of information is an important element in decision-making and forms the basis for the preparation of non-financial information and ESG (environment, social, governance) reports.

It should also be noted that the purpose of the study whose results are presented in this article is not to assess financial condition or performance, but rather the quality of the information that, regardless of its nature (whether it is financial or non-financial), shapes and influences communication between business entities and their stakeholders. Improving the quality, including comparability, of information is an essential element of the modern economy and a prerequisite for achieving sustainable development goals. It is also important to add that financial information and non-financial information are not separate areas but form an overall system of generating information that should be understandable, accessible, and comparable. This provides an opportunity to create a transparent flow of information between entities and stakeholders.

It should also be noted that this article, as well as the study, relates directly to the insurance sector, which is unique in that insurance companies are entities whose activities are determined to a large extent by the need to create a positive image for stakeholders. Insurers are entities of public trust and their products, due to their specific nature, determine the need to pay particular attention to sustainability objectives.

This paper is designed to assess the comparability of financial information presented in the annual statements of insurance companies by means of the NiCE (Nijmegen Centre for Economics) index of financial report quality assessment and suggest some directions for changes to the assessment of comparability of insurance companies' financial reporting.

Our methods are based on the NiCE index of financial report quality assessment described by Beest et al. [7]. Similar studies have been undertaken by, among others [2,6], though they did not address insurance companies. The NiCE research methodology is formalized and has a rating scale and described assessment rules, which ensures comparability of results. It is very important in making qualitative assessments, which are very difficult to carry out and have a lot of relative (discretionary) elements.

This article consists of four main parts. The first section is a review of literature on the comparability of information presented in a financial statement, the second part presents materials and methods, the third part the results of our research, and the final section (discussion) compares our results to those of similar research and offers some conclusions.

This investigation improves the state of the art in the assessment of financial reporting quality and fills a gap in the verification of comparability of insurance companies' financial information. In addition to filling a gap in the literature, the research carried out has implications for practice, as it highlights the importance of comparability of financial information for business development and maintaining a good image for the insurance company. Research is also important to insurance companies' sustainability challenges as it assesses the quality of financial information that underpins ESG reports.

## 2. The Comparability of Information Presented in a Financial Statement—A Review of Specialist Literature

Financial statements drafted by businesses are the best source of information about their performance and asset and capital condition. Isolated figures in a financial statement have no meaning in themselves and only become meaningful once compared to other figures. The comparison only makes sense when they are somehow connected to one another. Thus, data in financial reporting form information about the condition of a business and become useful to their recipients. To become useful, though, they should be collected and presented in line with specific principles. The principle of consistency, or comparability, is one of them. Accordingly, accounting policies adopted should be applied on a continuous basis when grouping economic transactions; valuing assets, equity and liabilities, appreciation, or redemption write-offs; calculating financial results; and compiling financial statements in such a way that the information arising from the statements over successive financial years is comparable. The principle of consistency is also expressed in disclosing the closing balances of assets, equity, and liabilities in a given financial year at the same amounts as their opening balances for the next year. This means the closing balances of the particular accounts of the main and subsidiary ledgers of a given reporting period should comply with the opening balances for the subsequent period.

The issue of the comparability of financial information as a major characteristic of financial reporting has long been discussed in the literature [8]. There are results that stress the role of comparability and its advantage over other qualities of financial statements [9–14]. The comparability of data involves an entity preserving a system and method of grouping transactions shown in the accounts and financial statement in successive reporting periods of financial years. Comparability analysis involves assessing data for the same company from different periods or between companies using different criteria for grouping them. This helps users to identify similarities and differences between two sets of economic phenomena.

The comparability of financial information affects many aspects of economic entities. Majeed and Chao [15] argue that greater comparability increases financial transparency, which improves the information environment and increases the liquidity of the stock. According to Chauhan and Kumar [16], comparability of financial statements is a unique qualitative characteristic of accounting information that enables investors to identify and understand similarities and differences between financial statements. Similar arguments are advanced by Chen and Gong [17], too, who point out the comparability is positively linked with the accuracy and precision of managerial forecasts, which means the comparability improves managers' ability to anticipate the future performance of a business. Habib et al. [18] claim a greater degree of comparability cuts the cost of obtaining information, reduces the uncertainty of result assessment, and increases the overall quantity of information available to those outside an undertaking, which in turn helps to remedy restrictions to outside financing.

The comparability of financial statements is often studied with reference to IASs/IFRSs (International Accounting Standards, International Financial Reporting Standards), which

contribute to its improvement. The introduction of IAS/IFRS standardises the financial reporting of businesses, leading to an enhanced standard and quality of the information disclosed and helps decision-makers to better understand the financial statements of competitors. This is corroborated by research by Almehairi et al., Jibril, Young and Zeng, Cascino and Gassen, and Yurisandi and Puspitasri, among others [2–6].

It should be pointed out, however, that the study by Conaway [19] indicates the enterprises applying local standards become more comparable to entities using the IFRS.

In the view of Schipper [20], the comparability of financial reporting attained via the adoption of IAS/IFRS helps to improve the availability of information to decision-makers by helping them understand financial reporting better and thus improves information transfers among enterprises and countries.

Barth et al. [21] claim the adoption of IASs/IFRSs and the international coordination of accounting regulations have improved the global comparability of book-keeping information.

Healy et al., Leuz and Verrechia, and Daske et al. [22–24] show the adoption of IASs/IFRSs has helped reduce the asymmetry of information by enhancing the quality of financial reporting. This is upheld by Almehairi et al. [2], who studied 20 companies tracked from 2015 to 2018. Their methods are based on the index of financial report quality assessment developed by NiCE. The research demonstrates the characteristics of book-keeping information, namely significance, faithful presentation, comprehensibility, and comparability, have improved substantially following the adoption of IASs/IFRSs.

In line with IAS 1: *Presentation of financial statements*, the target of comparability of data included in financial statements of an entity refers to its financial statements from earlier periods and to those of other entities. Thus, in accordance with the IAS/IFRS, the principle of comparability is realised in two respects:

(1)    As comparability over time—an entity's financials from different periods are compared,
(2)    As comparability in space—the financials of various economic entities are compared across sectors, regions, or countries.

Gierusz and Martyniuk [25] believe comparability with information disclosed within a given financial statement should additionally be addressed, defined as the internal coherence of a statement. This is implemented if the same methods of presentation are applied to similar assets and financial items.

It should be pointed out the comparability of information reported for subsequent periods cannot mean a ban on new, improved solutions. An entity may replace an existing solution as long as it serves the purpose of a clear and reliable presentation of its standing. Reasons for these changes must then be identified, their quantitative impact on the financial result must be identified, and the comparability must be assured of figures in the financial statement for the year prior to the year such changes are made.

Comparability in space, or between entities, requires the elimination of differences in accounting regulations, not only those prevailing in various countries but also applicable to businesses in a given country, and a limitation of choices by reducing the number of variant solutions acceptable under existing regulations.

It is essential that information in a financial statement be presented clearly. According to IAS 1: *Presentation of financial statements* [26], "Information is obscured if it is communicated in a way that would have a similar effect for primary users of financial statements to omitting or misstating that information. The following are examples of circumstances that may result in material information being obscured:

(1)    information regarding a material item, transaction or other event is disclosed in the financial statements but the language used is vague or unclear;
(2)    information regarding a material item, transaction or other event is scattered throughout the financial statements;
(3)    dissimilar items, transactions or other events are inappropriately aggregated;
(4)    similar items, transactions or other events are inappropriately disaggregated; and

(5)  the understandability of the financial statements is reduced as a result of material information being hidden by immaterial information to the extent that a primary user is unable to determine what information is material".

An analysis of IAS 1: *Presentation of financial statements* shows the principle of comparability requires a clarity of information in a financial statement, which involves presenting information in a clear, unambiguous, aggregated, and logical manner as far as the method of presentation and the significance of a given item to the assessment of an entity's financial standing are concerned. Thus, the principle of comparability comprises a substantial share of qualitative features, which interferes with its accurate preservation.

The application of the comparability principle implies the need to inform financial statement users of:

- The accounting principles applied
- Any changes to these principles
- Consequences of changes to the principles

According to IAS 1: *Presentation of financial statements* [26], "An entity whose financial statements comply with IFRSs shall make an explicit and unreserved statement of such compliance... In virtually all circumstances, an entity achieves a fair presentation by compliance with applicable IFRSs. A fair presentation also requires an entity:

(1)  to select and apply accounting policies in accordance with IAS 8: *Accounting Policies, Changes in Accounting Estimates and Errors*. IAS 8 sets out a hierarchy of authoritative guidance that management considers in the absence of an IFRS that specifically applies to an item;

(2)  to present information, including accounting policies, in a manner that provides relevant, reliable, comparable, and understandable information;

(3)  to provide additional disclosures when compliance with the specific requirements in IFRSs is insufficient to enable users to understand the impact of particular transactions, other events and conditions on the entity's financial position and financial performance".

A review of specialist literature on the comparability of information in financial statements is summarised in Table 1.

**Table 1.** A list of key publications on the comparability of financial information.

| Author(s) | Publication Year | Title of the Paper | Main Conclusions |
|---|---|---|---|
| Schipper, K. [20] | 2003 | Principles-Based Accounting Standards | The adoption of IFRSs improves the comparability of financial statements and streamlines the decision-making process |
| van Beest, F., Braam, G., Boelens, S. [7] | 2009 | Qualities of Financial Reporting: measuring qualitative characteristics | Implementation of IFRSs improves the qualitative characteristics of financial statements. |
| Barth, M.E., Landsman, W.R., Lang, M., Williams, C. [21] | 2012 | Are IFRS-based and US GAAP-based accounting amounts comparable? | The adoption of IFRSs improves the comparability of financial statements. |
| Jayaraman, S., Verdi, R. [27] | 2013 | The effect of economic integration on accounting comparability: Evidence from the adoption of the euro | The adoption of IFRSs improves the comparability of financial statements. This is particularly evident in countries in the euro area. |
| Loannis, T., Dionysia, D. [28] | 2014 | Value relevance of IFRS mandatory disclosure requirements | The adoption of IFRSs improves the comparability of financial statements. IFRS and GAAP analysis. |
| Cascino, S., Gassen, J. [5] | 2015 | What drives the comparability effect of mandatory IFRS adoption? | The adoption of IFRSs improves the comparability of financial statements, but mainly in public companies. |

**Table 1.** *Cont.*

| Author(s) | Publication Year | Title of the Paper | Main Conclusions |
|---|---|---|---|
| Yurisandi, T., Puspitasari, E. [6] | 2015 | Financial Reporting Quality—Before and After IFRS Adoption Using NiCE Qualitative Characteristics Measurement | The adoption of IFRSs improves the quality of financial statements. Survey of companies listed on the Indonesian Stock Exchange with the highest market capitalization. |
| Habib, A., Hasan, MM., Al-Hadi, A. [18] | 2017 | Financial statement comparability and corporate cash holdings | Greater comparability of financial statements reduces the cost of obtaining information and eliminates restrictions on external funding. |
| Chauhan, Y., Kumar, S.B. [16] | 2019 | Does accounting comparability alleviate the informational disadvantage of foreign investors? | Consider if the comparability of financial statements reduces the cost of obtaining information via foreign investors, which in turn increases their investments in businesses showing greater book-keeping comparability. A weak external information environment and a low interest of analysts are likely to enhance the benefits of book-keeping comparability with reference to Indian firms. |
| Chen, A., Gong, J.J. [17] | 2019 | Accounting comparability, financial reporting quality, and the pricing of accruals | Comparability of financial statements improves the usefulness of financial information. |
| Almehairi, M.N., Ketait, A., AlQassim, R., Grassa, R. [2] | 2021 | Does IFRS adoption enhance the financial reporting quality of DFM listed companies? | Comparability is proven to be positively linked to the accuracy and precision of managerial forecasts, which means it improves managers' ability to anticipate future performance. |
| Majeed, M.A., Chao, Y. [15] | 2022 | Financial statement comparability and stock liquidity: evidence from China | The adoption of IFRSs improves the comparability of financial statements. |
| Conaway, J.K. [19] | 2022 | Has Global Financial Reporting Comparability Improved? | Comparability of financial statements affects share liquidity. However, the impact is only significant for non-state-owned companies. |

Source: The authors' compilation based on [2,5–7,15–21,27,28].

The literature review demonstrates the international regulations help to keep accounts in line with a standard concept and resolve any doubts while preserving the principle of comparability. Meeting the requirement of financial information comparability is prerequisite to the comprehensibility and reliability of financial statements, necessary for the decision-making process to be effective. The comparability of financial information is important both to entities preparing financial statements and to their environment since it is essential to maintain it across reporting periods and businesses. This characteristic seems important to any entity regardless of the scope (sector) or scale of their operations or the size of a business. It is also a key feature of insurance companies' reporting systems, with which they try to present themselves as public trust organizations. Insurance companies have accounting and financial reporting systems different from those of other entities due to the specific nature of their business, i.e., insurance [29–33]. This does not produce a different approach to the realisation of financial information comparability, which is a universal characteristic. For the purpose of the article, the following research hypothesis

was formulated: the financial statements of insurance companies are characterised by high comparability of financial information.

## 3. Materials and Methods

In the context of the foregoing discussion, an empirical study was undertaken to establish if insurance companies provide interested stakeholders with comparable financial statements. The choice of cases to study was guided by their market shares. Following an analysis of the insurance market, 8 insurance companies were chosen. All of them had a share in the EU market, measured by the value of assets, of more than 30% (35.06–32.54%). The selection of the study cases was intentional, not random. Three annual reporting periods, namely, 2019, 2020, and 2021, were chosen to examine changes over time. The choice of the reporting periods was determined by the time of research, that is, the most up-to-date available financial statements were selected. The 2022 financial statements were not yet available at the end of the study, i.e., in February 2023. In effect, the study sample consists of 24 annual statements, that is, 3 reports each from the 8 cases. Only annual financial statements were reviewed, without considering the information on solvency and financial standing (Solvency and Financial Condition Report—SFRC).

The insurance sector was chosen because of the authors' interest and experience in this sector, and also because they are special institutions distinguished by their specificity in terms of their activities, regulations, and products, as well as their role in the economy and society.

The financial reports of the following selected insurance groups were studied:

– Allianz Group (Minhen, Germany),
– Aviva plc (London, UK),
– The Axa Group (Paris, France),
– CNP Assurances (Issy-les-Moulineaux, France),
– Credit Agricole GRICOLE Assurances (CAA) (Paris, France),
– Generali Group (Trieste, Italy),
– Vienna Insurance Group (VIG) (Vienna, Austria),
– Prudential plc (London, UK).

All groups report their financial data in accordance with International Financial Reporting Standards. Their status in the insurance market and history of operation argued for their choice as well. They are established in the market, and half of them date back to the 19th century. Aviva has been in the market for the shortest time, only 23 years. With the exception of Prudential Group, the European market is the core area of operation of the insurance companies examined.

Table 2 lists the insurance groups selected with regard to their country of core operation, year of foundation, and legal grounds for their financial reports.

**Table 2.** The characteristics of the selected insurance groups.

| Name of Insurance Group | Country of Core Operation | Legal Grounds of Consolidated Financial Statement | Year of Group Foundation |
|---|---|---|---|
| Allianz | Germany | IFRS | 1890 |
| Aviva | UK | IRFS | 2000 |
| AXA | France | IRFS | 1946 |
| CAA | France | IFRS | 1948 |
| CNP | France | IFRS | 1959 |
| Generali | Italy | IFRS | 1831 |
| Prudential | UK/USA | IRFS | 1848 |
| VIG | Austria | IFRS | 1824 |

Source: The financial reports of insurance groups reviewed (These are published as Annual Reports on the websites of the insurers concerned).

To illustrate the market power of the insurance companies studied, their total assets during the years examined are presented in relation to the total assets of the European market. Since Prudential Group reports its financials in US dollars, these are converted into euro at the European Central Bank (ECB) rates of exchange (The ECB rates of exchange applied EBC: 31 December 2019→USD 1 = EUR 0.8902, 31 December 2020→USD 1 = EUR 0.8149, 31 December 2021→USD 1 = EUR 0.8829). The figures for the particular insurance groups are given on the basis of financial reports published by the groups. Aggregated figures follow the Solvency II S.02.01 model and have been published by the European Insurance and Occupational Pensions Authority (EIOPA). The global figures are converted into euro at the ECB rates of exchange at the reference date.

Table 3 contains the characteristics of the examined insurance groups in terms of assets held.

**Table 3.** The assets of insurance groups studied.

| Name of Insurance Group | Value of Assets (EUR m) | | |
|---|---|---|---|
| | **2019** | **2020** | **2021** |
| Allianz | 1,011,185 | 1,060,012 | 1,139,429 |
| Aviva | 460,043 | 479,857 | 358,474 |
| AXA | 780,878 | 804,589 | 775,491 |
| CAA | 18,306 | 18,648 | 18,478 |
| CNP | 440,366 | 442,540 | 483,002 |
| Generali | 514,574 | 544,710 | 586,225 |
| Prudential | 404,341 | 420,567 | 175,787 |
| VIG | 50,344 | 50,428 | 52,178 |
| Total assets of the insurance group | 3,680,037 | 3,821,351 | 4,058,064 |
| Total assets of the market | 10,495,934 | 11,742,384 | Data not available |
| The shares of the groups examined in the market (%) | 35.06 | 32.54 | Data not available |

Source: [34].

The shares of the groups in the European market ranged from 32.54 to 35.06% during the years reviewed. The share for 2021 could not be calculated, unfortunately, as aggregated figures were not available at the date of this study. The shares can be presumed to be similar, however, as the assets were comparable to the preceding years.

The method described by Beest et al. [7] was employed. The assessment of financial statement comparability by NiCE relied on an original qualitative index. It continued to be utilised by: Yurisandi and Puspitasari and Almehairi et al., among others [2,6]. Their research addressed enterprises other than financial institutions. NiCE developed a comprehensive index for the assessments of financial reporting quality based on five features evaluated via five-point Likert scale. The characteristics of accounting information disclosed refer to IASB and FASB requirements and include materiality, faithful representation, comprehensibility, comparability, and currency. With regard to the assessment of information comparability, both measures concerning the consistent application of the same accounting principles and procedures by a given firm in the individual periods and those concerning the comparability across entities in a single period are used. Table 4 contains the principles of assessing the comparability of financial information according to NiCE and the measurement methods suggested by Beest et al. [7]. The NICE method was chosen because it is the only method known from the literature to assess information quality. Qualitative assessments are very difficult to carry out and have a great many relative (discretionary) elements, which makes it difficult to compare the obtained assessments, both over time and between different entities. The NiCE research methodology is formalised and has a rating scale and described assessment rules, which ensures comparability of results. It is also a proven method, because it has been used in other studies for a different group of entities.

**Table 4.** Principles for assessing the comparability of financial information according to NiCE.

| Number | Explanation | Explanation of Ratings |
|---|---|---|
| C1 | The notes to changes in accounting policies explain the implications of a change | 1 = Changes not explained<br>2 = Minimum explanation<br>3 = Explained why<br>4 = Explained why + consequences<br>5 = No changes or comprehensive explanation |
| C2 | The notes to revisions in accounting estimates and judgments explain the implications of a revision | 1 = Revision without notes<br>2 = Revision with few notes<br>3 = No revision/clear notes<br>4 = Clear notes + implications (past)<br>5 = Comprehensive notes |
| C3 | The company's previous accounting period's figures are adjusted for the effect of the implementation of a change in accounting policy or revisions in accounting estimates | 1 = No adjustments<br>2 = Described adjustments<br>3 = Actual adjustments (one year)<br>4 = 2 years<br>5 = >2 years + notes |
| C4 | The results of current accounting period are compared with results in previous accounting periods | 1 = No comparison<br>2 = Only with previous year<br>3 = With 5 years<br>4 = 5 years + description of implications<br>5 = 10 years + description of implications |
| C5 | Information in the annual statement is comparable to information provided by other organizations | Judgment based on: accounting policies; structure; explanation of events. In other words: an overall conclusion of comparability compared to annual statement of other organisations<br>1 = No elements<br>2 = 1–2 elements<br>3 = 3–5 elements<br>4 = 6–10 elements<br>5 = >10 elements |
| C6 | The annual statement presents financial index numbers and ratios | 1 = No ratios<br>2 = 1–2 ratios<br>3 = 3–5 ratios<br>4 = 6–10 ratios<br>5 = >10 ratios |

Source: The authors' compilation based on: Beest et al. [7].

C1 serves to assess the method and scope of presenting information about any accounting policy changes on the part of the entities studied. Determining the level of this presentation was partly subjective, since drawing lines between the particular levels requires some additional criteria, not always fully unambiguous.

C2 addresses the comparability of information about estimated changes and the presentation of their effects, or the comparability of financial statement contents. The assessment of this factor consisted of the identification of information about estimated changes in a financial statement, verifying the scope of its presentation, and describing the degree of impact. The method of assessing this factor comprises some subjective elements concerning the scope of estimated changes presented and the extent of their impact.

C3 serves to assess whether data for a previous business period are adjusted for changes to accounting policies or for valuation. The examination of this factor consisted of comparing data included in financial statements and an analysis of the attached notes. The questions of changes to figures can be evaluated objectively. Value changes are not

always clearly described in notes; therefore, the assessment of this factor comprises a subjective element.

In turn, C4 relates to the comparability of financial statements, primarily with the preceding financial years. Earlier investigations [2,6,7] have not defined the elements to be taken into accounts: only general data, the structure of a financial statement, or the method of reporting. The amounts disclosed in a statement are assessed automatically, most commonly with reference to a previous year. This means a subjective assessment of the elements to be reviewed for factor C4. Further analysis ought to define in more detail the scope of elements to be assessed.

Factor C5 refers to the comparability of annual report structures among organisations, not to the contents of individual elements. Although the method of its assessment involves certain simplifications, it is a valuable contribution to the overall comparability of insurance companies' financial statements. We have improved the assessment developed by F. Beest et al. [7] with respect to C5. They failed to specify the measures concerning C5 and merely explained in general what elements it should assess across entities, without attributing the measures in detail as per the Likert scale. Since a standard is absent for comparisons whether given information is comparable to information from other organisations, we have developed an original method of assessment. We identified 16 elements to be defined in the structure of annual statements. This set relates to the specific nature of the entities studied and we believe it should constitute the criterion of similar structures of insurance companies' annual statements. To maintain a certain cohesion of assessment between factors C5 and C6, the same values were adopted for the purposes of assessment, that is, 1 = no elements, 2 = 1–2 elements, 3 = 3–5 elements, 4 = 6–10 elements, 5 = >10 elements. The assessment is presented in Table 1, with the addition of the proposed method of assessing C5.

At the final stage of assessing the comparability of insurance companies' financial statements, the number of financial ratios cited by entities examined (factor C6) is taken into account. This assessment is fully measurable. Only the ratios relating to group studies are considered, without addressing the ratios concerning, for example, branches.

It should be stressed the assessment of 4 factors, namely, C1, C2, C3 and C4, is subjective in line with the method prepared by Beest et al. [7] and then adjusted by these authors. We approached the assessment with due diligence, yet we are aware its results are not free from the assessors' subjective angles.

Descriptive statistical methods served to develop the results. Basic statistical parameters and significance tests were developed. Statistica version 13.3 served to execute the calculations.

## 4. Results

Six factors (variables) are assumed to determine the level of comparability of financial information published in financial statements from insurance companies. Each factor is assessed along the five-degree scale (1–5—Likert scale). The basic statistics of the variables are set out in Table 5.

The results in Table 5 show the average for all the factors is quite high, above 4. This is further corroborated by the median, 4, for the first four factors (C1–C4) and 5 for the final two (C5–C6). The minimum values of the factors were quite varied, however. For C3, it was only 1, whereas for C5 and C6, it was as much as 4. It is different for the maximum values. They equalled 5 for all the six features. This confirms the excellent results of the insurance companies reviewed.

An analysis by means of variance shows a low variability of the results. The variance ranged from 0.23 for C6 to 1.21 for C3. The analysis of standard deviation yielded similar results. Again, the lowest value of 0.46 related to C6 and the highest, 1.10, to C3. These results mean the factors were similar across the group of insurance companies studied.

**Table 5.** The basic statistics of the variables studies.

| Specification | C1 | C2 | C3 | C4 | C5 | C6 |
|---|---|---|---|---|---|---|
| Average | 4.33 | 4.00 | 4.00 | 4.16 | 4.58 | 4.66 |
| Median | 4.00 | 4.00 | 4.00 | 4.00 | 5.00 | 5.00 |
| Minimum | 3.00 | 2.00 | 1.00 | 3.00 | 4.00 | 4.00 |
| Maximum | 5.00 | 5.00 | 5.00 | 5.00 | 5.00 | 5.00 |
| Variance | 0.41 | 1.13 | 1.21 | 0.40 | 0.25 | 0.23 |
| Standard deviation | 0.63 | 1.06 | 1.10 | 0.63 | 0.50 | 0.48 |
| Coefficient of variation | 14.70 | 26.58 | 27.58 | 15.28 | 10.98 | 10.31 |

Source: The authors' own research based on STATISTICA 13.3.

The coefficient of variation, the final row in Table 5, is rather low. This also signifies the factors are not statistically significant. The coefficient ranges from 10.31 to 27.58. This affirms that the results show little variation for C5 and C6, with the greatest differences for C2 and C3.

It can be concluded, therefore, the entities examined show similarities as far as the publication of financial information is concerned. It can be posited at this stage the financial statements reviewed are comparable.

Table 6 contains the results for correlations among the factors (variables) studied.

**Table 6.** The correlation matrix among the variables examined.

| Specification | C1 | C2 | C3 | C4 | C5 | C6 |
|---|---|---|---|---|---|---|
| C1 | 1.000000 | 0.385164 | 0.618590 | 0.178571 | 0.451754 | −0.047246 |
| C2 | 0.385164 | 1.000000 | 0.667124 | 0.641941 | 0.324799 | −0.254762 |
| C3 | 0.618590 | 0.667124 | 1.000000 | 0.618590 | 0.156492 | −0.245495 |
| C4 | 0.178571 | 0.641941 | 0.618590 | 1.000000 | 0.225877 | −0.236228 |
| C5 | 0.451754 | 0.324799 | 0.156492 | 0.225877 | 1.000000 | −0.059761 |
| C6 | −0.047246 | −0.254762 | −0.245495 | −0.236228 | −0.059761 | 1.000000 |

Source: The authors' own research based on STATISTICA 13.3.

C6 and the remaining variables are negatively correlated, to the maximum of −0.25 with C2. The remaining variables are positively correlated. The closest relationship between the variables reached 0.4–0.6 of the correlation coefficients.

C1 is the most strongly related to the variables C3 and C5. This denotes a rather strong correlation between the comparability of information on accounting policy changes, on the resultant adjustments to results, and the way this information is disclosed by the remaining entities.

C2, which concerns the information about changes of and updates to values in additional notes, is most strongly related to the variables C3 and C4. This means the way this information is presented in the notes is similar to the manner of presenting information on adjustments in the statement. The rather high correlation with C4 means a quite high comparability over particular periods in time.

C3 is more correlated with as many as three variables, that is, C1, C2, and C4. The coefficient of correlation was approx. 0.6. This is an obvious result, given that C3 served to assess to what degree values were adjusted on foot of changes to accounting policies, while C1 and C2 determined how the information is disclosed in the notes. On the other hand, C4 helped to assess the comparability of information in recent years.

C4 is 0.6 and in addition correlated with C2. This means there is a connection between the comparability in time and disclosing the information on changes in value adjustments and revaluations in notes.

Our results suggest C5 was not in any way strongly related to the remaining factors.

Table 7 includes the results of statistical tests for independent variables (C2–C6) and dependent variable (C1) with regard to the reference value of 3. For each variable, the mean, standard deviation, the valid number of observations, standard error, and *t*-test results are given.

**Table 7.** Statistical tests.

| Specification | Specification | | | | | | | |
|---|---|---|---|---|---|---|---|---|
| | **Mean** | **St. Dev.** | **Valid** | **St. Error** | **Reference Constant** | **t** | **df** | **p** |
| C1 | 4.333333 | 0.637022 | 24 | 0.130032 | 3.00 | 10.25392 | 23 | 0.000000 |
| C2 | 4.000000 | 1.063219 | 24 | 0.217029 | 3.00 | 4.60769 | 23 | 0.000124 |
| C3 | 4.000000 | 1.103355 | 24 | 0.225221 | 3.00 | 4.44008 | 23 | 0.000188 |
| C4 | 4.166667 | 0.637022 | 24 | 0.130032 | 3.00 | 8.97218 | 23 | 0.000000 |
| C5 | 4.583333 | 0.503610 | 24 | 0.102799 | 3.00 | 15.40223 | 23 | 0.000000 |
| C6 | 4.666667 | 0.481543 | 24 | 0.098295 | 3.00 | 16.95582 | 23 | 0.000000 |

Source: The authors' own research based on STATISTICA 13.3.

The results in Table 7 help compare the mean values of each variable to the reference value and determine if they are statistically significant. All the variables except for C6 are significantly different than the reference value at the significance $p < 0.05$. This suggests C2–C5 affect the value of the dependent variable (C1), since they are significantly different from the latter's mean value. The mean value of C6 is not significantly different to the reference value, which means it has no impact on the dependent variable.

Table 8 illustrates the function's model based on multiple regression, with C1 the dependent variable.

**Table 8.** The results of multiple regression.

| *n* = 24 | A Summary of the Dependent Variable Regression: C1 (R = 0.79008490 $R^2$ = 0.62423414 Corr. $R^2$ = 0.51985474 F(5,18) = 5.9804 *p* | | | | | |
|---|---|---|---|---|---|---|
| | **Standard b** | **Standard Error** | **Coefficient b** | **Error of b** | **t(18)** | **p** |
| Free expr. | | | 1.143561 | 1.433504 | 0.79774 | 0.435421 |
| C2 | −0.043811 | 0.219144 | −0.026249 | 0.131299 | −0.19992 | 0.843784 |
| C3 | 0.843761 | 0.207306 | 0.487146 | 0.119688 | 4.07013 | 0.000718 |
| C4 | −0.392540 | 0.200640 | −0.392540 | 0.200640 | −1.95644 | 0.066112 |
| C5 | 0.427481 | 0.153599 | 0.540726 | 0.194288 | 2.78311 | 0.012273 |
| C6 | 0.081550 | 0.150615 | 0.107881 | 0.199245 | 0.54145 | 0.594838 |

Source: The authors' own research based on STATISTICA 13.3.

The results of C1 regression imply the model of linear regression for the data set explains about 62% of the dependent variable's variation (designated as $R^2$), a moderately good value. The corrected $R^2$, which addresses the number of variables in the model, is around 52%, a quite good result, too. F is 5.9804 and *p* is below 0.05, which suggests the regression model is statistically significant and has a considerable effect on the dependent variable, or C1. It needs to be pointed out that R (0.790) indicates a strong positive correlation between the independent variables and the dependent variable. The results of this regression suggest C1 is quite strongly connected with the independent variables C3 and C5 ($p = 0.000718$ and $p = 0.012273$, respectively), but less correlated with C2, C4, and C6 ($p = 0.843784$, $p = 0.066112$ and $p = 0.594838$, respectively).

## 5. Discussion

C1 was intended to determine the depth of detail of publication and explication of information about changes to accounting policies in additional notes. This assessment is largely subjective since not all of its levels are unambiguous. Even if level 1 (changes not explained) is easy to identify, the remaining levels (2–5) are not.

It should be added that the entities examined are insurance companies, large institutions obliged to publish extensive reports, often as integrated statements including financial statements. These institutions do not run into situations where information on changes to accounting policies is absent or merely mentioned without any additional explanations. Levels 1 and 2 do not apply, therefore.

All the insurance companies reviewed exhibit levels 3, 4, or 5 of the publication and explication of information on changes to accounting policies. The levels are varied with respect to the detail and extent of explaining of such changes as well as the clarity of this information concerning their consequences and impact on the entities' accounts. Nonetheless, the assessment was partly conditioned by how clear and complete the information is to users.

C2 relates to the comparability of information about estimated changes and the presentation of their effects. This is a crucial qualitative feature of the comparability of financial statements that affects a range of financial categories, including the value of assets and the financial result. The assessment of this factor involved the verification of the presence and extent of information about estimated changes, in particular, its impact on an insurer's financial position. Like in the case of other factors, some levels of the scale were not easy to identify. This is especially true of the following levels: 2—revision with few notes, 4—clear notes + implications (past), and 5—comprehensive notes. Two levels of the scale, i.e., 1—revision without notes and 3—no revision, were precise. Research should continue to specify the principles of assessing this factor, defining the scope of the notes to revisions in accounting estimates and judgements and their links to, for example, the financial result or the value of assets. The number and scope of references in a given financial statement should be added to the assessment scale, too. This would help to improve the quantitative as well as the qualitative features of the assessment (few notes, clear notes, comprehensive notes).

C3 addresses considering changes to accounting policies or the impact of adjustments on values disclosed in a financial statement. These issues could be noted in the data presented, since in some insurance companies, the data for a previous year are different than their values disclosed for a given financial year. These questions are clarified in additional notes. C3 relates primarily to results. This factor is hard to estimate by external observers, since data about these changes are quite cautiously published by insurance companies and result from individual accounting policies of a given company. Assessment 4—addressing the adjustments of the current and previous years with a brief description in the notes—prevails. Level 2 was awarded to one insurer, which provided legal grounds without any specific explanations. The assessments for this factor are subjective and dependent on the needs of statement users and knowledge of the subject. The scale would need to be more detailed to enhance the objectivity of a given assessment.

C4 refers to the comparability of the current accounting period results to those of previous financial periods. The comparability is preserved in most cases due to the prevailing regulations. C3, consideration of changes to accounting policies, interferes with the question. It compels statement users to seek additional data, e.g., in notes, to assess the dynamics of particular data. The preservation of the financial statement structure and units of data measurement published by a given insurance group over the years is to be applauded. This does not always maintain comparability among the individual insurance companies. The insurance companies report in diverse currencies (mostly euro and US dollars) and units (thousand, million, billion). The extent of comparability of reporting data needs to be more specific as far as this factor is concerned. The changes to regulations and the particular entities' accounting policies interfere with an objective assessment of

this factor. C4 is also reviewed by the management of insurance groups in order to enhance the positive tendencies in a given company, e.g., as regards increasing insurance premium. This means this assessment is highly dependent on the individual or entity undertaking the assessment.

An examination of factor C5 showed the criteria selected had largely been addressed in the structure of the entities' annual statements, which resulted in high assessments of this indicator (4–5). It should be emphasised, though, this assessment does reflect a full comparability of the institutions' annual reports. Although the criteria are addressed, their sequence and scopes vary. Some criteria form separate chapters of the structure, chapter subsections, or parts of a subsection.

The comparability of report structures between periods must be approved of. This is not addressed in the method of examination yet is an undoubted strength of the financial statements reviewed. Considerable similarities can be observed both with regard to criteria (some minor differences between the years were noted), the order and extent of discussion of the particular criteria.

An analysis of C6 also shows the method is not fault-proof. The financial statements of insurance companies were reviewed, and the number of ratios were calculated. The high numbers of ratios presented translates into a good assessment of C6 in all the entities studied. The ratios are chosen by the entities on an individual basis, however, which prevents their comparisons in space (the particular insurance companies present various ratios). Added to this, only few entities discussed the way they compute a given ratio, which also substantially undermines the comparability in space. The place of ratio presentation in the particular sections of financial statements (no standard approach to the parts where the ratios are presented and their scattering across the financial statements) greatly limits the comparability of this aspect. Like in the case of factor C5, the comparability between reporting periods, not part of the method in place, should be commended. The entities largely presented the same financial ratios in the same sections of their annual reports in the successive years.

Compared to earlier published results, this study of the insurance sector (as represented by several largest insurers in Europe) yields somewhat better results. This means that the research hypothesis formulated in the article, that the financial statements of insurance companies are characterised by high comparability of financial information, has been positively verified. The basis for positive verification is the review of research presented above and the conclusions obtained by the authors of this article. A survey by Beest et al. [7] reported an average comparability ranging from 3.07 (for C6) to 3.93 (C5). Yurisandi Puspitasari [6] reported highly varied levels, on introducing the IFRSs ranging from 2.38 for C4 to 4.7 for C5. In the most recent research [2] in the last of the years examined, 2018, the comparability averaged from 2.9 for C4 to 4.0 for C1. That study addressed not the narrow insurance market, but entities from a variety of sectors.

However, it should be emphasised here that a very important element of this research is that it relates to the insurance sector. Previous studies whose results have been published do not refer to insurance companies. In our opinion, all financial institutions and especially insurance companies should be examined separately, and the results obtained cannot be directly compared to other economic entities.

This article therefore fills a research gap and makes an important contribution to the economic science of the field of insurance in the context of sustainability.

Insurance companies, which are financial institutions, evidently display a greater comparability of financial information than other entities. This is corroborated by the present results. This is most likely caused by the far more detailed requirements of financial reporting and the nature of financial institutions itself, entities of public trust, which involves a greater transparency of the sector's information policies. This should be viewed positively in the context of future sustainability challenges. The high quality of financial information means that insurance companies are well prepared to produce ESG reports that will rely heavily on financial information. Sustainable development is also information that meets

certain characteristics of its usefulness. These include the comparability of information, which is particularly important in decision-making processes. The conducted research shows that insurance companies prepare information that meets the comparability feature. Therefore, one should be optimistic about the changes that will take place in insurance companies in connection with the implementation of the challenges of sustainable development and in particular with their information obligations in this regard—ESG reporting.

## 6. Conclusions

This paper was intended to assess the comparability of financial information in the annual statements of insurance companies. This purpose has been realised, since the specialist literature has been reviewed and a method has consequently been selected to carry out the study and formulate some conclusions to assess the comparability of financial information for a chosen sector.

Annual statements from the eight largest insurance companies in Europe in the period from 2019 to 2021 were reviewed using the method based on the NiCE index of financial report quality assessment described by Beest et al. [7]. This paper:

- Only focuses on the comparability of financial information (other studies have addressed other features of the information, such as materiality, faithful representation, comprehensibility, and currency);
- This study addresses one selected, specific industry, namely, insurance companies.
- The following key conclusions of this study can be identified:
- The factors used to assess the comparability are not statistically significant;
- The analysis of variability of the factors examined shows a minor variability of the results, which means the manner of presenting financial information by the insurance companies studied is similar;
- The overall level of comparability is quite high, at least 4 on a 5-point scale for all the factors;
- The results for the European insurance companies are somewhat higher, though not significantly different than those reported by other authors;
- The results of the regression function suggest the dependent variable C1 is quite strongly correlated with the independent variables C3 and C5, which means the manner and scope of presenting changes to accounting policies are rather strongly correlated with adjustments to accounting estimates as a result of changes from preceding periods and linked with the methods of presenting information in statements from the group of entities examined,
- Improvements to the model of assessing the comparability of insurance companies' financial statements in respect of factors C1–C6.

This study has helped to establish the financial information presented in the annual statements of the largest insurance companies in Europe is comparable. No substantial improvement in this respect could be noted between 2019 and 2021; however, the standards are similar in each of the years examined. It must of course be stressed that insurance companies are quite heavily regulated with regard to financial reporting, which means their reports are normally on a higher qualitative level than of other, especially smaller entities. This should be viewed positively in the context of the future challenges of reporting for sustainability purposes—ESG reporting.

The method employed seems not to be ideal. The assessment of some factors is partly subjective, which may distort the results to some extent. Using the method to assess the comparability of non-financial information (ESG information), which insurance companies have been bound to publish for some time, may prove an added challenge and a subject of future research.

**Author Contributions:** Conceptualization, M.C.-L., M.L. and K.B.; methodology, M.C.-L.; software, M.L.; formal analysis, M.C.-L., K.B. and E.S.; investigation, M.C.-L. and K.B.; resources, M.L. and K.B.; writing—original draft, M.L. All authors have read and agreed to the published version of the manuscript.

**Funding:** The publication was co-financed from the subsidy granted the Cracow University of Economics—WAP 2023. The article is co-financed by the Casimir Pulaski Radom University as part of the program DBUPB/2018/009/3378/181/P.

**Institutional Review Board Statement:** Not applicable.

**Informed Consent Statement:** Not applicable.

**Data Availability Statement:** Not applicable.

**Conflicts of Interest:** The authors declare no conflict of interest.

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
