# Peer review of "The Comparability of Financial Information in Insurance Companies Using NiCE Qualitative Characteristics Measurement"

_sustainability, doi:10.3390/su152014828_

Round 1

Reviewer 1 Report

Authors focus on comparability of financial information by usage of NiCE index. They are strongly involved in calculation, and they do not conclude why such comparisons are needed, who is beneficiary of such comparisons, how that comparisons impact on compared companies.

Authors are requested to explain all the above issues.

Authors apply for publishing in Sustainability Journal Economic and Business Aspects of Sustainability

Unfortunately they did not write about sustainability. Hence, the paper should be changed or it should be submitted to a journal which collect papers on financial information.

Sorry, As I see Authors are not interested in sustainability.

The literature survey is very poor. Unfortunately, as authors want to submit research paper, they should provide professional literature survey.

They should explain the results of similar studies in other countries. May be in other countries , the similar studies on comparisons of financial or insurance institutions were done.

The research methodology presentation must be improved. Authors are requested to formulate research questions and hypotheses.

All acronyms are to be explained at their first usage (e.g., IAS/IFRS)

Table 1

“What drives the comparability effect of        - the question is of what ?

Next row

“mandatory IFRS adoption?”            - why ? , very unclear

“Do IFRS Adoption Enhance the Financial “ – what financial, very unclear

Table 2 source

“The financial reports of insurance groups reviewed.”

Please, give the names of that reports

In this paper authors focus on applicability of NiCE index , instead on research.

Minor editing of English language required

Author Response

Dear Reviewer,

Thank you very much for your review, your time and all your comments and suggestions. We have improved our article according to the comments in the review, in particular:

  • We have pointed out the relationship between the content of our article and sustainability,
  • We have explained the need for comparability of information, why it is important, for whom and how it affects entities in the insurance sector.
  • We formulated a hypothesis,
  • We have tried to enriched the literature review, and checked one more time all bases but we have not found anything more; we would like to point out that the small amount of this literature is also an indicator of the research gap and our take on this topic,
  • We have explained the acronyms,
  • We have referred to results from other previous studies available in the literature when describing the results, but our article is focused on insurance companies. This is a very important element of our study. We did not find in any available studies with scientific publications the results of the research relating directly to insurance companies,
  • We also made the other corrections mentioned in the review (e.g., we corrected Table 1, etc.)

Thank you again for the review, which gave us the opportunity to improve our article and contributed to its scientific value.

kind regards

The Authors

Reviewer 2 Report

I have carefully read your manuscript "The comparability of financial information in insurance companies using NiCE qualitative characteristics measurement” and I enjoyed it. 

I think the manuscript is discreetly written and structured, addressing an interesting and important topic.         

However, I believe that to accept the paper, the authors still need to focus on numerous and significant changes and justify choices underlying the research.

In my opinion, the abstract should - represent, albeit in a nutshell, the academic, managerial, and institutional implications.   

The structure of the paper must be inserted in the last part of the "Introduction" paragraph.

The literature review in the "Introduction" paragraph could be moved to a stand-alone paragraph (e.g. paragraph 2).

In the paragraph “Materials and Methods:

The logic behind the choice of the insurance industry should be more explained and supported.

The rationale behind the choice of sample should be explained and supported.

The rationale behind the choice of time frame should be explained and supported.

The rationale behind the choice of NiCE should be explained and supported.

The discussion of the results should be strengthened with reference to the results highlighted by previous studies comparable in whole or in part with those of research under consideration.

The contribution in terms of originality and innovativeness of the authors should be more explicit.

Managerial, academic, and institutional implications should be more explicit.

All the best! And good luck with your research.

Minor editing of English language required

Author Response

Dear Reviewer,

Thank you very much for your review, your time and all your comments and suggestions. We have improved our article according to the comments in the review, in particular:

  • we have corrected the abstract,
  • we have completed the information on the structure of the article in the introduction,
  • we have included a separate section for the literature review,
  • we explained why we chose the insurance sector,
  • we have completed the remaining points when describing the method and assumptions of the study,
  • we have referred to previous research findings,
  • we have added a description of our contribution to the study,
  • we have added information on how our research findings are relevant to management, the insurance sector, and academic methods.

Thank you again for the review, which gave us the opportunity to improve our article and contributed to its scientific value.

kind regards

The Authors

Reviewer 3 Report

The paper could benefit from an introductory paragraph that targets the purpose of the paper and the methodological process behind the used method, in this case the NiCE solution.

The paper could also benefit from a paragraph that highlights the data collection process and what are the direct and indirect limits of the model.

Furthermore, the authors could insert a paragraph in the Discussion section that shares the issues had finalising the research and also future paths that could result from the actual research.

Author Response

Dear Reviewer,

Thank you very much for your review, your time and all your comments and suggestions. We have improved our article according to the comments in the review, in particular:

  • we have completed the introductory information to the article related to the purpose and application of the NICE method,
  • we have completed the remaining issues in the description of the method and study design,
  • we also completed the information on future studies.

Thank you again for the review, which gave us the opportunity to improve our article and contributed to its scientific value.

kind regards

The Authors

Round 2

Reviewer 1 Report

The paper was improved , however there are still some weaknesses. 

The paper is an application of method, hence that method of evaluation should be strongly justified. Authors just mentioned about sustainability, but they should strongly emphasize the impact of findings on sustainability. 

They have done a literature survey, But as I see literature survey is fundamental research method, hence, authors are requested to expand literature reference list. 

The proof reading is necessary. 

yes, proof reading is needed 

Author Response

Dear Reviewer,

Thank you very much for your review, your time and all your comments and suggestions. We have improved our article according to the comments in the review.

We would like to further clarify the link between our research and sustainability. The starting point for our research is the assumption that financial reporting and non-financial reporting are closely linked. They are not two separate areas, as used to be treated, but form a coherent system for generating information for stakeholders. In particular, this coherence is extremely important in the context of the activities of insurance companies, which are financial institutions, and the assessment of their financial condition is an integral part of their information policy. A sustainable market is not only about environmental and climate aspects, but also about respecting human rights, so transparency of information to all stakeholders is equally about treating groups such as customers, companies using insurance, reinsurers, banks, supervisory institutions or ordinary people and their families equally. All of them should have access to transparent, readable, and understandable information about an insurer's financial performance, which is a determinant of its position in today's sustainability market. And this is what our research is for.

Thank you again for the review, which gave us the opportunity to improve our article and contributed to its scientific value.

kind regards

The Authors

Reviewer 2 Report

Dear Author(s),

I'm happy with the revisions made. 

Thanks a lot for answering to all my queries in a separate file. It was useful to assess the strength of the paper.

I think you followed all my advice and suggestions.

Best Regards

Author Response

Dear Reviewer,

Thank you very much for your review.

kind regards

The Authors